# Left ventricular mass normalization for body size in children based on an allometrically adjusted ratio is as accurate as normalization based on the centile curves method

Hubert Krysztofiak [1,2]*, Marcel Młyńczak [3], Łukasz A. Małek[4], Andrzej Folga[2], Wojciech Braksator [5]

**1** Mossakowski Medical Research Centre, Polish Academy of Sciences, Warsaw, Poland, **2** National Centre for Sports Medicine, Warsaw, Poland, **3** Warsaw University of Technology, Faculty of Mechatronics, Institute of Metrology and Biomedical Engineering, Warsaw, Poland, **4** Department of Epidemiology, Cardiovascular Disease Prevention and Health Promotion, National Institute of Cardiology, Warsaw, Poland, **5** Department of Sports Cardiology and Noninvasive Cardiovascular Imaging, 2nd Medical Faculty, Medical University of Warsaw, Warsaw, Poland

* hkrysztofiak@imdik.pan.pl

## Abstract

### Background

Normalization for body size is required for reliable left ventricular mass (LVM) evaluation, especially in children due to the large variability of body size. In clinical practice, the allometrically adjusted ratio of LVM to height raised to the power of 2.7 is often used. However, studies presenting normative LVM data for children recommend centile curves as optimal for the development of normative data. This study aimed to assess whether the allometrically adjusted LVM-to-height ratio can reliably reproduce the results of LVM normalization for height based on the centile curves method.

### Methods

Left ventricular mass was computed for 464 boys and 327 girls, 5–18 years old, based on echocardiographic examination. Normalized data representing LVM for height were developed using the centile curves construction method and two variants of the allometrically adjusted ratio method: one variant with the allometric exponents specific to the study groups, and one variant with the universal exponent of 2.7. The agreement between the allometric methods and the centile curves method was analyzed using the concordance correlation coefficient, sensitivity, and specificity.

### Results

For both the specific allometric variant and the universal variant, the analysis of concordance has indicated high reproducibility compared to the centile curves method. The respective coefficient values were 0.9917 and 0.9916 for girls, and 0.9886 and 0.9869 for

**Data Availability Statement:** All relevant data are within the manuscript and its Supporting Information files.

**Funding:** The authors received no specific funding for this work.

**Competing interests:** The authors have declared that no competing interests exist.

boys. The sensitivity and specificity test has also shown high agreement. However, for girls, the sensitivity was higher for the specific variant (100% vs. 90.9%).

## Conclusion

The results of the study show that allometric scaling of LVM for height can very reliably reproduce the results of LVM normalization for height based on the centile curves method. However, the analysis of sensitivity and specificity indicates greater agreement for the allometric normalization with the group-specific allometric exponents.

## Introduction

Echocardiographic linear heart dimensions and left ventricular mass (LVM), which is calculated based on the linear dimensions of the left ventricle [1,2], are examined in daily clinical practice. The calculation of LVM allows confirmation of the presence of left ventricular hypertrophy (LVH) [2], which is a concern in cardiovascular diseases and a predictor of poor outcome [3,4]. It is also a concern in athletes in whom regular exercise causes physiological changes to the heart, including hypertrophy [5,6,7]. In athletes, a proper LVM assessment and differentiation of physiological LVH from pathological is essential. Normalization is required for reliable LVM evaluation because the size of the heart varies with the size of the body. LVM normalization is of particular importance in children and adolescents due to the large variability of body size in children of similar age.

Normalization of cardiac size for body size is a standard approach in clinical evaluation and recommended by the guidelines [2,8,9,10]. However, there is a discussion about the best body size variable and scaling methodology for cardiac size normalization [11–17].

In a recent study, we showed that among the frequently used body size variables, i.e., body surface area (BSA), lean body mass calculated based on predictive equations, and height, only height provides reliable normalization of LVM [11]. This study is a continuation of our previous work. It addresses the issue of scaling methodology.

The simplest methods of LVM scaling in clinical settings are based on the ratio of LVM to the body size variable (LVM index). The most popular is the simple ratio of LVM and BSA [18], and the ratio of LVM divided by height raised to the power of 2.7 [12]. The latter is an allometrically adjusted ratio and is based on a simple allometric model of a relationship between LVM and height. The value of 2.7 is a commonly used exponent related to the curvilinearity of the relationship (allometric exponent). The LVM index is easy to calculate, and the result can be presented as a numerical value or as a standardized value (z-score). More advanced methods are based on the constructing LVM centile curves for body size, the same as the well-known growth curves of height-for-age, weight-for-age, and weight-for-height used for the pediatric population [19]. A normalized LVM can be presented as a standardized value (z-score) or visualized on a centile chart. It is considered that the centile curves method is more accurate than the ratio methods for normalization of LVM for body size [15–17] and therefore allows for a more reliable diagnosis of LVH. However, the methodology of constructing centile curves, and then computing standardized LVM values, is complicated; that is why this method is not widely used in clinical practice. Additionally, since only a few studies have evaluated an agreement between the ratio method and the centile curves method [15–17], the superiority of the latter method is not confirmed. Should we then strive to replace the ratio method of LVM normalization with the more sophisticated method of centile curves in clinical practice?

We have attempted to provide a substantive basis to answer this question and designed a study to examine the concordance between the centile curves method and the allometrically adjusted ratio method. The study aimed to assess whether the allometrically adjusted LVM-to-height ratio can reliably reproduce the results of LVM normalization for height based on the centile curves method.

## Materials and methods

### The study group

It was a retrospective study based on data derived during periodic medical evaluation of child and adolescent athletes. All of the studied children were engaged in regular athletic training at the local or national level (mainly soccer, track and field, basketball, swimming, and martial arts). The entire study group consisted of 791 healthy White children (327 girls and 464 boys), aged from 5 to 18 years. All the study participants underwent transthoracic echocardiography as a part of medical evaluation because of innocent heart murmurs or suspicion of abnormal electrocardiographic findings. The athletes in whom echocardiography revealed significant acquired or congenital heart diseases, affecting normal heart size and hemodynamics, were not included in the study. The anthropometric measures, i.e., height and body mass, were measured during the main examination.

### Echocardiography

Echocardiographic examinations were performed by two experienced sonographers using a commercially available ultrasound scanner (Toshiba Aplio 400, Toshiba Medical Systems Europe, Zoetermeer, the Netherlands), according to recent guidelines. All measurements were taken in the 2-dimensional parasternal long axis view (PLAX) at end-diastole and included the basic linear cardiac dimensions necessary for LVM computing: left ventricular internal dimension (LVIDd), interventricular septal thickness (IVSd), and posterior wall thickness (PWTd). All measurements were taken from the inner edge to inner edge and reported to within 1 mm. Left ventricular mass was computed according to the formula of Devereux RB et al. [1]:

$$LVM = 0.8\{1.04[(LVIDd + PWTd + IVSd)^3 - (LVIDd)^3]\} + 0.6$$

### Ethical considerations

The Ethics Committee of the Medical University of Warsaw approved the study procedure (approval AKBE/75/17). As the study was retrospective, and the data used were collected during routine medical monitoring, neither written nor verbal consent was required for this particular study. However, each subject, or the subject's parent or guardian, had signed the informed consent form for the routine medical monitoring, including a statement of agreement to the use of the results for scientific purposes.

### Development of left ventricular mass normative data

Since previous studies about LVM normalization in children and adolescents have indicated that LVM normative data should be sex-specific [15,17,20], all analyses were carried out separately for girls and boys. Thus the entire study group was divided into two sex-specific Study Groups. Three sets of LVM-for-height normative data were developed based on each group's records, using three different methods of normalization. First, the LMS method was used to construct centile curves [21]. In this method, based on the relationship between LVM and height in the study group, the expected mean LVM (M), coefficient of variation (S), and skewness (L) for each height level are generated. For an individual child, the LVM z-score is then

calculated from the L, M, and S values corresponding to the child's height, according to the equation:

$$z-score = \frac{\left[\left(\frac{actual\ LVM}{M}\right)^L - 1\right]}{L \times S}$$

Next, the allometrically adjusted ratio method was used. In this method, the height is raised to the power equal to the exponent from the power law equation (allometric equation) describing the curvilinear relationship of LVM with height [12,22]. Two variants of the method were tested. In the first variant (height$^b$), allometric equations specific to our Study Groups were fitted for the bivariate relationship of LVM with height, and sex-specific allometric exponents were determined. These equations have the general form $LVM = a(body\ size)^b$, where $b$ is the allometric exponent. Logarithmic transformation gives the linearized form of this equation (ln($LVM$) = ln($a$)+$b$ln($height$)), allowing estimation of the allometric coefficients using linear least squares regression modeling [22]. The sex-specific allometric exponents were used to transform height, which is used as a denominator in the ratio method. Then, for each subject, LVM was divided by transformed height. Thus, new variables of indexed LVM were produced, and normative data expressed as a mean and standard deviation of the LVM indexes developed. Next, LVM z-scores were calculated, according to the equation:

$$z-score = \frac{\left(\frac{actual\ LVM}{height^b}\right) - \left(mean_{normative\ data\ \frac{LVM}{height^b}}\right)}{standard\ deviation_{normative\ data}}$$

In the second variant (height$^{2.7}$) of the allometrically adjusted ratio method, the universal allometric exponent 2.7 was used to develop normative LVM-for-height data for both sex-specific Study Groups. Like in the first variant, normative data expressed as a mean and standard deviation were produced, and z-scores were calculated with the equation analogous to that in the first variant.

For proper normalization of LVM for body size, it is necessary to eliminate body size information from the normalized LVM [22]. To check whether the body size information had been eliminated in the produced normative data, we tested whether there was a relationship between the calculated LVM z-scores and height. The Pearson correlation coefficient and the slope of the linear regression line for each set of the LVM z-scores were examined, and graphical presentations of the data were inspected.

## Comparison of different methods of the LVM normalization

In this part of the study, from each of the sex-specific Study Group, 200 subjects were randomly assigned to corresponding Test Groups, to assess the concordance of different LVM normalization methods. For this comparison, following the aim of this study, we assumed that the LMS method is a reference method for LVM normalization. The z-scores calculated based on the LVM normative data obtained according to the allometrically adjusted ratio methods were compared to those calculated based on the L, M, and S values from the LMS method [21]. This allowed us to evaluate the reproducibility of the allometric methods and to assess their sensitivity and specificity compared to the LMS method.

At first, we examined whether the mean differences between the z-scores calculated based on both allometric LVM normative data and those calculated based on the LMS normative data differ from 0. The paired sample t-test was used. The allometric z-scores were then plotted against the LMS z-scores on scatter graphs. Regression lines were fitted to the data, and equality lines were drawn. For the initial assessment of whether the allometric LVM normalization

methods can reproduce the results of the LMS method, the Pearson correlation coefficients and linear regression coefficients, as well as the slopes and y-intercepts, were estimated for the various data sets.

Next, the concordance correlation coefficient (CCC), introduced by Lin LI [23], was used to evaluate the agreement between the normalization methods. The CCC converts the mean squared difference between the paired points of two data sets into a correlation coefficient that measures how far the corresponding data points deviate from the equality line (y = x) in a two-dimensional coordinate system. The CCC is a product of accuracy and precision. The measure of precision is the Pearson correlation coefficient, which measures how far the points deviate from the best-fit line. A measure of accuracy is the coefficient that measures how far the best-fit line deviates from the equality line. This coefficient is considered as the bias correction factor and depends on the location shift and scale shift. A CCC value of 1 represents perfect agreement, a value of minus 1 represents perfect disagreement, and a value of 0 represents no agreement. The equation for calculating the concordance correlation coefficient, as well as equations for the intermediate factors, are presented in the supplementary S1 Table.

Finally, the sensitivity and specificity of the allometric methods, in comparison to the LMS method, were evaluated for the entire study group of 791 children and adolescents. For this analysis, the subjects were classified as having LVH when their z-score > 1.65 [15].

All calculations were performed using R (version 3.5.2, "Eggshell Igloo"; R Foundation, Vienna, Austria, http://www.r-project.org), along with external packages, especially the gamlss package (version 5.0), which contains a function used to fit the LMS curves, and the DescTools package (version 0.99.28), which contains tools used to estimate the concordance correlation coefficients. For all statistical tests, a significance level of $\alpha = 0.05$ was used.

## Results

### Subjects characteristics

The characteristics of the entire study group, the sex-specific Study Groups, and the sex-specific Test Groups are presented in Table 1. The sex-specific Study Groups were used for the development of the LVM normative data. The Test Groups, both consisting of subjects randomly selected from the corresponding Study Groups, were used to analyze the agreement between the LVM normalization methods. The entire study group of 791 children and adolescents was used to evaluate the sensitivity and specificity.

### The LVM normative data for the mutual comparison

Two sets of LVM-for-height normative data, separate for girls and boys and generated based on the LMS method, are provided as L, M, and S values in supplementary text files (S1 and S2 Datasets, respectively). The allometric exponents estimated for the sex-specific Study Groups are presented in Table 2. The LVM normative data computed based on the LVM-to-height ratio are expressed as means and standard deviations and also included in Table 2. For the height$^b$ variant, the height is raised to the power of $b$, where $b$ is equal to the estimated allometric exponent; for the height$^{2.7}$ variant, the height is raised to the power of 2.7.

Based on the developed LVM normative data, the LVM z-scores were calculated, according to the LMS method of normalization and according to both variants of the allometric method. Examples of LVM z-score calculations are presented in a supplementary file (S1 Text). The Pearson correlation coefficients and the slopes of the linear regression lines of the relationships between the calculated LVM z-scores and heights are also presented in Table 2. All the coefficients are non-significant. In addition, graphic presentations of the data show point configurations that do not indicate the presence of non-linear relationships (S1 Fig). Thus, both variants

**Table 1. Characteristics of the entire study group, the sex-specific study groups, and test groups.**

| | Entire study group | Study Group | Test Group | Study Group | Test Group |
|---|---|---|---|---|---|
| | | Girls | Girls | Boys | Boys |
| Number of subjects | 791 | 327 | 200 | 464 | 200 |
| Age [years] | 12 (5–18) | 12 (5–18) | 11 (6–18) | 13 (5–18) | 13 (6–18) |
| Height [cm] | 157 (111–194) | 153 (111–188) | 153 (117–181) | 162 (112–194) | 167 (112–194) |
| Body mass [kg] | 45.4 (18.2–100.0) | 41.8 (18.8–86.1) | 40.9 (20.0–84.7) | 48.9 (18.2–100) | 53.5 (18.2–97.6) |
| LVM [g] | 104.50 (38.77–280.47) | 93.46 (38.77–213.18) | 93.25 (38.77–180.13) | 114.13 (45.91–280.47) | 122.67 (46.52–263.05) |
| LVIDd [mm] | 44 (31–60) | 42 (34–55) | 42 (34–54) | 45 (31–60) | 46 (31–60) |
| IVSd [mm] | 8 (5–13) | 8 (5–11) | 7 (5–11) | 8 (5–13) | 8 (5–12) |
| PWTd [mm] | 8 (5–13) | 7 (5–10) | 7 (5–10) | 8 (5–13) | 8 (5–12) |
| Training volume [minutes] | 270 (60–630) | 240 (60–630) | 240 (60–540) | 270 (60–630) | 300 (60–630) |
| Resting HR [beats/minute] | 71 (45–93) | 75 (49–93) | 76 (51–93) | 68 (45–93) | 69 (47–93) |
| Systolic BP [mmHg] | 114 (80–135) | 110 (80–135) | 109 (80–135) | 116 (86–135) | 117 (88–135) |
| Diastolic BP [mmHg] | 64 (40–85) | 64 (40–85) | 63 (40–85) | 65 (40–85) | 65 (40–85) |

Data are expressed as "median (minimum–maximum)"; LVM, left ventricular mass; LVIDd, left ventricular internal dimension; IVSd, interventricular septal thickness; PWTd, posterior wall thickness; training volume is a measure of participation in sports activity and was estimated as the product of the average number of training sessions per week and the average duration of a session; HR, heart rate; BP, blood pressure.

of the allometric LVM normalization method, as well as the LMS method, eliminated the height information from the normalized LVM.

## The agreement between the LVM normalization methods

For girls, the mean and standard deviation (in parentheses) of LVM z-scores were 0.0791 (1.0122) for the LMS normative data, 0.0748 (1.0121) for allometric LVM normative data produced based on the specific allometric exponent, and 0.0750 (1.0090) for allometric LVM normative data based on the exponent of 2.7. For boys, these were 0.0213 (1.0297), 0.0282

**Table 2. The LVM normative data for allometrically adjusted LVM-to-height ratios and the parameters used to assess the relationship between the normalized LVM and height for the LMS method and both variants of the allometric method.**

| | LMS | height$^b$ | height$^{2.7}$ |
|---|---|---|---|
| **Girls** | | | |
| Allometric exponent | N/A | 2.5848 | 2.7 |
| LVM to height ratio | N/A | 32.0467 (5.1431) | 30.5527 (4.9191) |
| Pearson's coefficient | 0.0015 (ns) | 0.0000 (ns) | -0.0719 (ns) |
| Slope of regression line | 0.0001 (ns) | 0.0000 (ns) | -0.0048 (ns) |
| **Boys** | | | |
| Allometric exponent | N/A | 2.8118 | 2.7 |
| LVM-to-height ratio | N/A | 32.5524 (6.1043) | 34.2606 (6.4542) |
| Pearson's coefficient | 0.0004 (ns) | 0.0127 (ns) | 0.0864 (ns) |
| Slope of regression line | 0.0000 (ns) | 0.0007 (ns) | 0.0045 (ns) |

Here, "LVM-to-height ratio" refers to the LVM normative data computed based on the allometrically adjusted LVM-to-height ratio and is expressed as "mean (standard deviation)." For height$^b$, the height is raised to the power of $b$, where $b$ is equal to the estimated allometric exponent; for height$^{2.7}$, the height is raised to the power of 2.7. The Pearson correlation coefficient and the slope of the linear regression line both correspond to the relationship between the calculated LVM z-scores and height. "ns" stands for "non-significant" ($p \geq 0.05$).

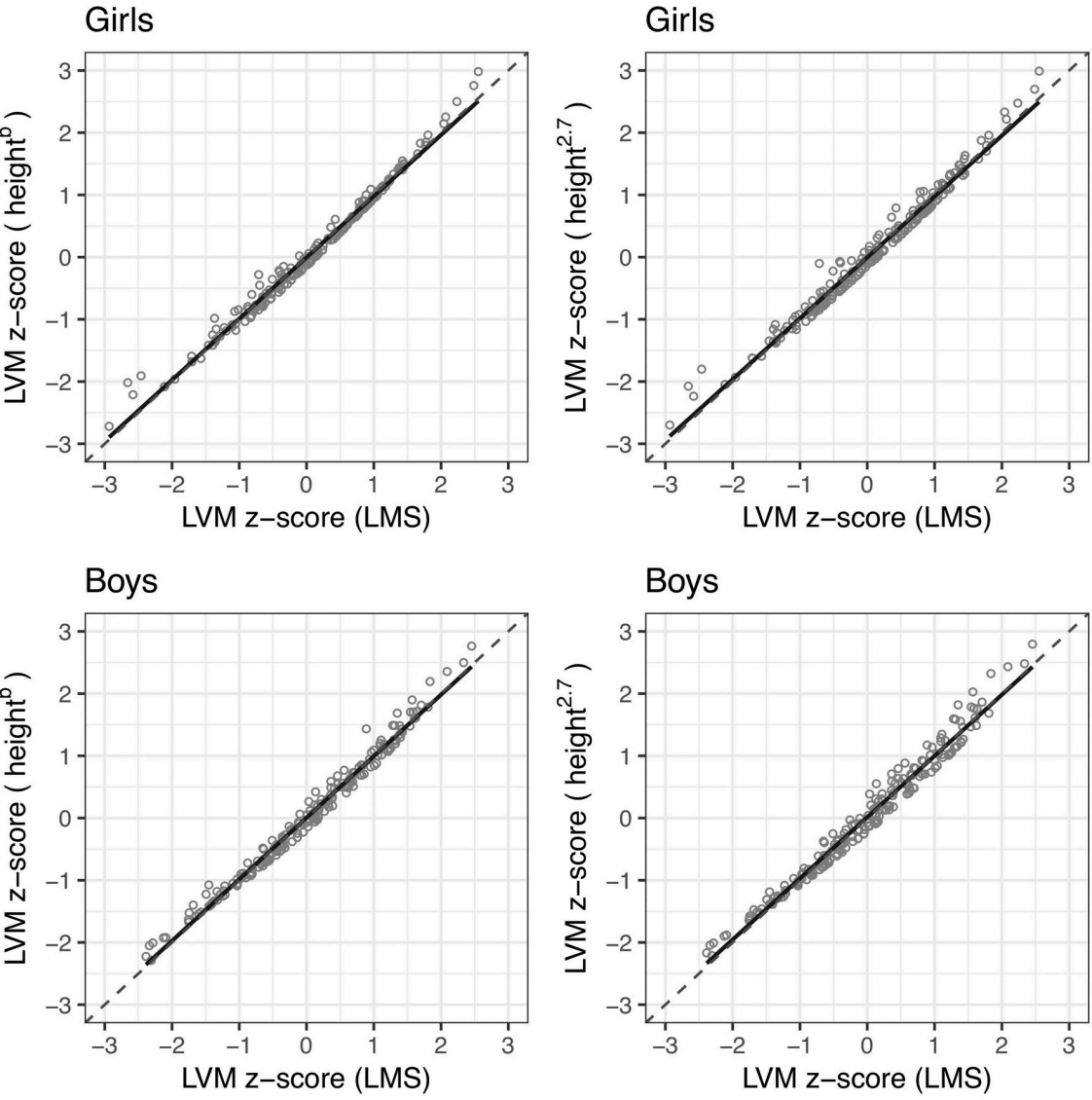

**Fig 1. Scatter plots of the LVM z-scores calculated based on the allometric normative data against the LVM z-scores calculated based on the LMS normative data.** On each chart, the regression line is fitted to the data points (solid line), and the equality line is shown (dashed line). The charts on the right correspond to the height$^b$ variant and those on the left to the height$^{2.7}$ variant. The upper charts are for girls and the lower for boys.

(1.0340), and 0.0378 (1.0344), respectively. The mean differences between the z-scores calculated based on both allometric LVM normative data sets and based on the LMS normative data did not differ from 0, for both female subjects and male subjects.

The height$^b$ and height$^{2.7}$ allometric LVM z-sores, separate for girls and boys, are plotted against the LMS z-score in Fig 1. These scatter graphs show that, for both variants, data points are congregated along the equality line, and the best fit line overlaps the equality line.

Table 3 presents the results of statistical analyses that test the agreement between the allometric normalization methods and the LMS method. The coefficients of the regression lines indicate that the best-fit lines are very close to the equality line. The y-intercepts of the best-fit lines are close to 0 and are non-significant (the lines pass through the origin). The slopes of the lines are close to 1 and are significant, as are the Pearson correlation coefficients.

**Table 3. Evaluation of the agreement between the allometrically adjusted ratio methods of LVM normalization and the LMS method.**

| | height$^{b}$ | height$^{2.7}$ |
|---|---|---|
| **Girls** | | |
| Mean squared difference | 0.0169 | 0.0231 |
| Slope of regression line | 0.9918 (p<0.001) | 0.9916 (p<0.001) |
| Y-intercept of regression line | 0.0049 (ns) | 0.0047 (ns) |
| Pearson correlation coefficient | 0.9917 (p<0.001) | 0.9886 (p<0.001) |
| Bias correction factor | 1.0000 | 1.0000 |
| **Concordance correlation coefficient (CCC)** | **0.9917** | **0.9886** |
| Lower one-sided 95% confidence limit for CCC | 0.9895 | 0.9857 |
| Scale shift | 1.0001 | 1.0031 |
| Location shift | 0.0043 | 0.0041 |
| **Boys** | | |
| Mean squared difference | 0.0178 | 0.0279 |
| Slope of regression line | 0.9874 (p<0.001) | 0.9825 (p<0.001) |
| Y-intercept of regression line | -0.0066 (ns) | -0.0159 (ns) |
| Pearson correlation coefficient | 0.9916 (p<0.001) | 0.9870 (p<0.001) |
| Bias correction factor | 1.0000 | 0.9999 |
| **Concordance correlation coefficient (CCC)** | **0.9916** | **0.9869** |
| Lower one-sided 95% confidence limit for CCC | 0.9894 | 0.9834 |
| Scale shift | 0.9958 | 0.9954 |
| Location shift | 0.0067 | 0.0160 |

height$^{b}$ and heigh$^{2.7}$ stand for variants of the allometric normalization method that have been compared with the LMS method. In the height$^{b}$ variant, sex-specific allometric exponents (represented as *b* in the allometric equation) were estimated based on the relationship between LVM and height in our Study Groups. In the heigh$^{2.7}$ variant, a universal allometric exponent 2.7 was used to develop sex-specific normative data for LVM. "ns" means "non-significant" (p $\geq$ 0.05).

The concordance correlation coefficients reflect the initial evaluation. For the height$^{b}$ normalization, for both girls and boys, CCC is equal to the Pearson correlation coefficient because the bias correction factor is equal to 1. The situation is similar for the height$^{2.7}$ normalization. Location shifts and scale shifts are minimal. The lower one-sided 95% confidence limits for all the CCCs are high.

The sensitivity and specificity of the allometric methods, compared to the LMS method, and the distribution of negative and positive LVH classifications according to the allometric normalization methods, compared to the LMS method, are shown in Table 4.

## Discussion

The results of our study show clearly that the allometrically adjusted LVM-to-height ratio normalization method can very reliably reproduce the results of LVM normalization for height based on the centile curves method. Thus, there is no reason to replace the allometric normalization of LVM with a seemingly-better procedure of centile curves construction like the LMS method.

In this study, we assumed that the LMS method for constructing normalized data, as an advanced statistical method [15–17,21], is the most accurate, and can be regarded as the reference method. We compared standardized results from two variants of the allometric LVM normalization to the results of the LMS method. In the first allometric variant, sex-specific

**Table 4. Evaluation of sensitivity and specificity of different variants of the allometric method of LVM normalization in comparison to the LMS method.**

|  | height[b] | height[2.7] |
|---|---|---|
| Sample size | 791 | 791 |
| Number of true positives | 31 | 30 |
| Number of true negatives | 751 | 751 |
| Number of false positives | 9 | 9 |
| Number of false negatives | 0 | 1 |
| Sensitivity | 100.00% | 96.77% |
| 95% Confidence Interval for Sensitivity | 88.78% - 100.00% | 83.30% - 99.92% |
| Specificity | 98.82% | 98.82% |
| 95% Confidence Interval for Specificity | 97.76% - 99.46% | 97.76% - 99.46% |

As in Table 3, height[b] and heigh[2.7] stand for variants of the allometric normalization method that have been compared with the LMS method. The subjects were classified as having LVH when their LVM z-score > 1.65. Confidence intervals for sensitivity and specificity are Clopper-Pearson exact confidence intervals.

allometric exponents were modeled based on the LVM and height data from our Study Groups of young athletes. In the second variant, we used the universal allometric exponent of 2.7. For both these variants, the normative data development and agreement analyses were performed separately for girls and boys.

It seems that there is not much difference between allometric normalization using a specific allometric exponent as compared to the universal allometric exponent. However, the exact numbers are slightly better for the specific exponents, and analysis of sensitivity and specificity also indicates the allometric normalization with the specific allometric exponents as the preferred method.

## Important questions about the cardiac size scaling procedure

An estimation of LVM is necessary for the diagnosis and management of left ventricular hypertrophy [2,6]. Athletic training can be responsible for LVH [24,25], though athleticism does not preclude the influence of pathological factors. Sometimes, it is difficult to differentiate physiological hypertrophy from pathological [5,7].

There is no doubt that proper LVM evaluation, or more generally, cardiac dimensions evaluation, requires normalization for body size [9]. This need is particularly evident in children and adolescents, whose heart size changes with age, in parallel to the size of the body, but also in the youth of the same age, where large variations in body size are observed. In athletes, especially speed and strength athletes, muscle mass significantly increases as a result of a specific exercise regime, with parallel changes in LVM [26,27]. A question thus arises: what body size variable is the best for LVM normalization? In our recent study, we indicated that among commonly used scaling variables, like BSA, LBM computed based on the predictive equation, and height, only height allows for reliable evaluation of LVM in child and adolescents athletes [11]. This is because normalized LVM is underestimated after normalization against BSA or computed LBM in subjects with high body mass, like speed and strength athletes, and overweight and obese subjects.

In this study, we address another question about the best method of LVM normalization, since there is a gap between clinical practice and some recommendations from echocardiographic studies. In clinical practice, the most popular LVM normalization methods are the simple (linear) ratio of LVM divided by BSA and the allometrically adjusted ratio of LVM

divided by height raised to the power of 2.7. The guidelines recommend these methods of LVM scaling [8]. In turn, in studies presenting LVM normative data for children and adolescents, the development of LVM normative data is most often based on advanced statistical methods of constructing normalized data, like the LMS method, introduced for the construction of growth curves [15–17]. Some researchers recommend this method as more accurate [15–17,22,28]. Another question then arises: are there significant differences in accuracy and precision between the LVM normalization methods recommended for clinical use by guidelines, and the advanced procedures used in studies developing normalized LVM data?

It can be assumed that the LMS procedure is very accurate for LVM normalization because this method fits data points very well. The echocardiographic studies where the LMS method was used recommended this method for LVH diagnosis, indicating its better performance compared to the ratio methods [15–17]. In these studies, the LMS method was regarded as the reference, and different body size variables were used for LVM normalization of the compared data. It must be noted that because of the lack of a procedure that is able to determine or exclude LV hypertrophy with certainty, any conclusions based on the comparisons between different methods, favoring one method or disqualifying another, are questionable. Furthermore, the different body size variables used for LVM normalization further biased the results of these studies. Our study was the first attempt to directly compare the agreement between the LMS method and the allometric ratio method for LVM normalization, with the same body size variable used as a descriptor. This study has indicated that the allometric LVM normalization provides reproducible results compared to the LMS normalization.

## An optimal exponent for allometric LVM normalization for height

Allometry, a term introduced by Julian Huxley and George Tessier in 1936 [29], means "different measures" and refers to the differences in growth rate between parts of the body or between organs and body as a whole during development. This phenomenon is often observed in nature [30]. A comparison of an organ's size changes with the body's size changes shows that the course of the changes is nonlinear. However, a specific feature of allometric growth is that, after a bivariate logarithmic transformation, this course of changes becomes linear, and the relationship can be described using a linear equation [22,30]. The slope of the line is a measure of the relative growth of the organ with body size; it is referred to as the allometric coefficient, also known as the allometric exponent. This is because, for original data, the power law equation describes the course of changes.

Cardiac size changes with body size during development, and when one analyzes how LVM changes with body mass, the allometric exponent estimated for different groups is close to 1 [30,31]. Since both the variables are geometrically similar, the relationship is actually linear. For the relationship between LVM and height, the allometric exponent is estimated from about 2.4 to 3.2, depending on the group tested [12,32]. This relationship is allometric, and the estimated values of the exponent are consistent in terms of the dimensions of the two variables because a relationship between a three-dimensional variable and a one-dimensional is modeled.

Allometry is applied to scale LVM to height. Most often, the ratio of LVM to height is used where the height is adjusted with the allometric exponent [12,32,33]. In clinical practice, the allometric exponent of 2.7 is used [8]. This exponent was introduced by de Simone G et al. n 1992 [12] based on a study of two different groups of adults and one group of children and adolescents. In this study, different exponents were computed for different groups, but 2.7 was the exponent estimated for the pooled group.

Our study examines two variants of the allometric method of LVM normalization for height: with the allometric exponent of 2.7 as well as with specific exponents estimated for the sex-specific Study Groups. It was not the aim of our study to question the exponent recommended by de Simone G et al. [12]; therefore, we did not make a direct comparison between those variants. However, since our Study Groups consisted of child and adolescents athletes, we were interested in whether the application of specific allometric exponents would improve the performance of the allometric method when testing against the centile curves method of LVM normalization.

Although the exact estimates of reproducibility indicate slightly better agreement in the case of the specific allometric exponents, the graphs and the concordance coefficients reflect high precision and accuracy, with minimum location shift and scale shift, for both the allometric variants. However, the sensitivity of the specific variant is again slightly better. Thus, a question arises: during the development of LVM normative data with the allometric method, should specific allometric exponents first be estimated, or the universal exponent accepted as reliable?

This universal exponent of 2.7 has been challenged by others [32,33]. For example, 1.7 was recommended as more accurate for LVM quantification based on a study on two large cohorts of middle-aged and older adults of varying ethnicity [32]. In this study, instead of bivariate modeling, multivariate was applied by accounting for sex. As a major drawback of 2.7, researchers have indicated the presence of a significant negative relationship between normalized LVM and height. Additionally, the researchers stated that LVH defined based on the newly recommended ratio "was most consistently associated with cardiovascular events and all-cause death." This latter opinion was disputed by de Simone G and Devereux RB [34].

The problem of a relationship between normalized LVM and height after scaling LVM for height to the power of 2.7 was also raised in a study of 400 White children [33]. The study group consisted of infants, children, and adolescents of both sexes, aged 0–18. Researchers modeled allometric equations specific to sex-specific sub-groups and the pooled group and searched for the ratio adjustment that eliminated this relationship. They stated that a ratio of LVM to height raised to the power of 2.16 with a correction factor of 0.09 (height$^{2.16}$ + 0.09) has better reliability for diagnosing LVH in children. This variant of allometric scaling looks like a statistical exercise and has not been applied in clinical practice. However, this study, as well as the former, stimulates doubts about a universal exponent for allometric LVM scaling, and its reliability across sexes or age groups. The problem started with the study of de Simone et al. [12] where the biological context of allometry was omitted. The bivariate modeling of LVM against height for a pooled group of children and adults did not consider potential differences in the allometric relationships between these groups; ontogenetic (developmental) allometry vs. static allometry. The next problem is the drive for the universality of the LVM scaling procedure. One allometric exponent for all: males, females, children, adults, and special groups like athletes or infants. Is it necessary? Is it feasible?

To answer those questions was not the aim of our study. For sure, a special consensus is needed in this field. We sought to answer the question of whether the allometric scaling of LVM for height can reliably reproduce the results of the advanced but more complicated normalization based on the centile curves method. However, parallel analysis with two variants of the allometric method, with the universal exponent and with exponents specific to young athletes, has let us estimate the performance of both variants. The specific variant seems to work better, but the universal variant is almost equally effective. Most importantly, in our study, for a group of child and adolescent athletes aged 5–18, we did not find a relationship between normalized LVM and height in the variant with specific exponents, nor in the variant with the universal exponent.

## Study limitations

Our study has limitations. We used a group of child and adolescent athletes from 5 to 18 years of age for this analysis; the group was ethnically homogenous. It might be argued that such a characteristic of the group limits the possibility of generalization. We do not question the necessity of further research to confirm the results in younger children, adults, and subjects from different ethnic groups. However, because we call for group-specific exponents in allometric scaling of LVM, the characteristic of the group in this study is consistent with this postulate.

We used height as the normalizing variable for both the allometric method and the centile curves construction method. A question may raise of why we did not use BSA, recommended by the guidelines, or LBM, suggested as the optimal body size variable for LVM normalization. The selection of a normalizing variable is a consequence of our previous study of bias related to body mass, which is introduced to the normalized LVM by BSA or LBM computed based on predictive equations [11]. There, we show that when BSA or equation-based LBM is used for normalization of LVM, the normalized LVM is underestimated in subjects with a high body mass index. The analysis indicates that only the height-based normalization of LVM is free of the BMI bias.

The LVM normative data, generated using both the LMS method and two variants of the allometric procedure, were developed based on the sex-specific Study Groups. Subjects in the Test Groups that were used for the evaluation of the agreement between the methods of LVM normalization were randomly selected from the respective Study Groups. The lack of a distinct group for the assessment of the agreement may be considered a limitation. However, since this research included the development of LVM normative data with different methods, and this procedure requires large samples, all the participants were assigned to the Study Groups. The reproducibility analysis based on the Test Group that consisted of randomly selected subjects from the Study Group is statistically valid. The number of subjects in each Test Group was large enough for this analysis.

The Bland-Altman method is the most commonly used method to measure agreement [35] and a question may arise why we did not use this method. The Bland-Altman plot is not suitable for our data; the plot has a limitation when the outer tails of the compared data distributions are divergent. In our data, the extreme data points of both, the LMS z-scores and the allometric z-scores, contribute to such divergence. As a result, the differences between the z-scores are greater on the outer ends. When this situation occurs, the distribution of the differences is not normal. The Bland-Altman analysis requires normal distribution; the measurement variables need not be normally distributed, yet their differences should be. An additional limitation is the measurement range from negative, through zero, to positive. According to the general idea of Bland-Altman analysis, if the assumption of normality is not met, data may be logarithmically transformed. In the presented case, they cannot, because the z-scores are positive and negative. Besides, an analysis directed to a search for the proportional bias is not practicable. The Bland-Altman plot shows a biphasic configuration of data points and even a parabolic shape. The range of the measurement also limits ratio calculation: there is a risk of dividing by 0. The ratios estimated for z-scores close to 0 might be immense, and may highly influence the calculation, causing misinterpretation of the results.

## Conclusions

The results of the study show that in children and adolescents from 5 to 18 years of age, the allometric scaling of LVM for height very accurately reproduces the results of a more advanced centile curves construction procedure represented by the LMS method. Both allometric

variants, one with an exponent specific to the tested group, the other with the universal exponent of 2.7, meet the statistical criterion of efficacy: the normalized LVM does not show a linear relationship with the height.

In agreement analysis, the difference between the LVM index with an allometric exponent that is specific to a tested group and the LVM index with the universally used exponent of 2.7 is small, yet the former provides better concordance with the LMS method. Besides, the sensitivity and specificity evaluation, with the LMS procedure as the reference method, indicates a better sensitivity of the LVM index with the specific exponent compared to the universal LVM index; the specificity is similar.

For reliable clinical evaluation of LVM in school-aged children and adolescents, there is no need to replace the allometrically adjusted LVM-to-height ratio with the more sophisticated method of centile curves. However, it seems that group-specific allometric exponents should be used to avoid constraints related to incomplete elimination of body size information from the normalized LVM, and for better performance in daily clinical practice.

## Supporting information

**S1 Dataset. The L, M, and S values of the LMS method for girls.**
(TXT)

**S2 Dataset. The L, M, and S values of the LMS method for boys.**
(TXT)

**S3 Dataset. The original dataset.**
(TXT)

**S1 Fig. Scatter graphs of LVM z-scores against height.**
(TIFF)

**S1 Table. The equation for calculating Lin's concordance correlation coefficient and equations for the intermediate factors.**
(DOCX)

**S1 Text. Examples of LVM z-score calculations.**
(DOCX)

## Acknowledgments

The authors thank Martin Berka for linguistic adjustments.

## Author Contributions

**Conceptualization:** Hubert Krysztofiak, Marcel Młyńczak.

**Data curation:** Hubert Krysztofiak, Marcel Młyńczak, Łukasz A. Małek, Andrzej Folga, Wojciech Braksator.

**Formal analysis:** Hubert Krysztofiak, Marcel Młyńczak, Łukasz A. Małek.

**Funding acquisition:** Hubert Krysztofiak.

**Investigation:** Hubert Krysztofiak, Łukasz A. Małek, Andrzej Folga, Wojciech Braksator.

**Methodology:** Hubert Krysztofiak, Marcel Młyńczak, Łukasz A. Małek.

**Project administration:** Hubert Krysztofiak.

**Resources:** Hubert Krysztofiak.

**Software:** Hubert Krysztofiak, Marcel Młyńczak.

**Supervision:** Hubert Krysztofiak.

**Validation:** Hubert Krysztofiak, Marcel Młyńczak, Łukasz A. Małek.

**Visualization:** Hubert Krysztofiak, Marcel Młyńczak.

**Writing – original draft:** Hubert Krysztofiak.

**Writing – review & editing:** Hubert Krysztofiak, Marcel Młyńczak, Łukasz A. Małek, Andrzej Folga, Wojciech Braksator.

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
