## [Decision Letter · Decision Letter 0]

13 Aug 2019

PONE-D-19-17289

Left ventricular mass normalization for body size in children based on an allometrically adjusted ratio is as accurate as normalization based on the centile curves method.

PLOS ONE

Dear HUBERT KRYSZTOFIAK

Thank you for submitting your manuscript to PLOS ONE. After careful consideration, we feel that it has merit but does not fully meet PLOS ONE’s publication criteria as it currently stands. Therefore, we invite you to submit a revised version of the manuscript that addresses the points raised during the review process.

We would appreciate receiving your revised manuscript by Sep 27 2019 11:59PM. To enhance the reproducibility of your results, we recommend that if applicable you deposit your laboratory protocols in protocols.io, where a protocol can be assigned its own identifier (DOI) such that it can be cited independently in the future. For instructions see: http://journals.plos.org/plosone/s/submission-guidelines#loc-laboratory-protocols

We look forward to receiving your revised manuscript.

Kind regards,

Alejandro Diaz

Academic Editor

PLOS ONE

2. Please provide additional details regarding participant consent. In the ethics statement in the Methods and online submission information, please state whether you obtained consent from parents or guardians. If the need for consent was waived by the ethics committee, please include this information.

Additional Editor Comments (if provided):

I have read with great interest the work of Hubert Krysztofiak and collaborators entitled "Left ventricular mass normalization for body size in children based on an allometrically adjusted ratio is as accurate as normalization based on the centile curves method."

The authors clearly present the hypothesis, the objectives and analysis of the data. The writing of the article is clear and correct.

The exact definition of left ventricular hypertrophy in pediatrics is one of the most investigated and still unsolved items. The original contribution of the present study is essential to approximate positions between the different groups that are dedicated to the investigation of this issue.

In my opinion as an editor it is a very interesting study that deserves to be considered for publication in Plos One (MINOR CHANGES).

Reviewers' comments:

Reviewer's Responses to Questions

**Comments to the Author**

1. Is the manuscript technically sound, and do the data support the conclusions?

Reviewer #1: No

Reviewer #2: Yes

Reviewer #3: Yes

2. Has the statistical analysis been performed appropriately and rigorously? 

Reviewer #1: No

Reviewer #2: Yes

Reviewer #3: Yes

3. Have the authors made all data underlying the findings in their manuscript fully available?

Reviewer #1: Yes

Reviewer #2: No

Reviewer #3: Yes

4. Is the manuscript presented in an intelligible fashion and written in standard English?

Reviewer #1: Yes

Reviewer #2: Yes

Reviewer #3: Yes

5. Review Comments to the Author

Reviewer #1: I have evaluated with interest the manuscript "Left ventricular mass normalization for body size in children based on an allometrically adjusted ratio is as accurate as normalization based on the centile curves method.

The authors mistakenly use the guidelines developed for adults as a bibliographic reference (Lang 2015) without considering the recommendations for children and adolescents (Lopez 2010, Lopez 2018). It is known that in pediatrics specific percentiles based on sex and age should be used to know the evolution of growth. The authors provide an approach to normalize ventricular mass and avoid standardization based on percentiles but do not fail to show the advantages of this approach.

Reviewer #2: The article addresses a very relevant topic for the area of pediatric medicine (cardiovascular area). The writing is very clear, and the analyzes performed are explained in detail. The authors are clearly aware of the problem analyzed. I have some suggestions and requirements that I believe can increase the quality of the manuscript.

Page 6: "Three sets of LVM-for-height normative data were developed based on each group’s records, using three different methods of normalization". From which table, database and/or web were the values of LVM (M), coefficient of variation (S), and skewness (L) for each height level obtained, to calculate the z-score for each child ? Please be specific. How were the "expected, normative or reference" mean values and standar deviation levels obtained (calculated), to calculate the z-score by the allometrically adjusted ratio method (two variants)? Considering that PlosOne is a Journal read by professionals from a wide range of disciplines, it would be important to give specific examples of these calculations, indicating where the values used come from (e.g. please consider including this in an Annex).

Page 8: "For proper normalization of LVM for body size, it is necessary to eliminate body size

information from the normalized LVM [20]. To check whether the body size information had

been eliminated in the produced normative data, we tested whether there was a relationship

between the calculated LVM z-scores and height. The Pearson correlation coefficient and the

slope of the linear regression line for each set of the LVM z-scores were examined". Why were only linear models used for this purpose?

Page 8: "Comparison of different methods of the LVM normalization: In this part of the study, from each of the sex-specific Study Group, 200 subjects were randomly assigned to corresponding Test Groups, to compare different LVM normalization methods".Why did you work with a subsample in this part of the analysis? As an example, did you consider using Bootstrapping?

Page 8: "The z-scores calculated based on the LVM normative data obtained according to the allometrically adjusted ratio methods were compared to those calculated based on the L, M, and S values from the LMS method [19]. This allowed us to evaluate the reproducibility of the allometric methods and to assess their sensitivity and specificity compared to the LMS method". Why not also use a Bland and Altman analysis for repeated samples? Do the differences between methods remain the same for any level of body-height (proportional errors)? It would be important to include this (very simple) analysis, which although it has limitations, is widely used to analyze the agreement between tools. Please add it.

Page 9: "For this analysis, the subjects in the Test Groups were classified as having LVH when their z-score>1.65". Add reference

Results (page 10): The authors mentioned in the methodology: " This was a retrospective study based on data derived during periodic medical evaluation of child and adolescent athletes". Table 1. Please include more information on the subjects, in order to know to what extent their values are within "normality" (healthy expeted values). Information on the clinical and / or hemodynamic characteristics (blood pressure, heart rate, other echocardiographic data) of the subjects should be included. In addition, please include the minimum, median and maximum value of each variable. It is important to know the levels of the variables for which the study results are applicable.

Results: The authors mentioned in the methodology: "The athletes in whom echocardiography revealed significant acquired or congenital heart diseases, affecting normal heart size and hemodynamics, were not included in the study". It would be interesting to know the results of a similar analysis performed in this subgroup of children and adolescents. You can do it? In this way it could be known to what extent the methods have similarity in these special cases.

Results: "The Pearson correlation coefficients and the slopes of the linear regression lines of the relationships between the calculated LVM z-scores and heights are also presented in Table 2". Please, can you show these graphics?

Results: Table 5 and Table S2. Only 10 cases (in 200) were "positive" for LVH. Do you consider this "n" appropriate for a sensitivity and specificity analysis? Can you report confidence intervals? Can you present this analysis for "the whole group" (for separate and non-separated sexes).

Discussion (page 16): "It seems that there is not much difference between allometric normalization using a specific allometric exponent as compared to the universal allometric exponent. However, the exact numbers are slightly better for the specific exponents, and analysis of sensitivity and specificity also indicates the allometric normalization with the especific allometric exponents as the preferred method". An important question that arises is: Are children studied similar (in terms of LVM and body height) to children from other places. Could the authors perform an analysis and / or discuss this aspect? It is important for the purpose of understanding how generalizable the results are. Please consider comparing yourself to a population of different latitudes. An example may be the following article:

• Díaz A et al. Reference Intervals and Percentile Curves of Echocardiographic Left Ventricular Mass, Relative Wall Thickness and Ejection Fraction in Healthy Children and Adolescents. Pediatr Cardiol. 2019 Feb;40(2):283-301.

Discussion: An interesting aspect would be to know to what extent the differences (although apparently not significant) between methods, could be explained by other co-factors (e.g. sex, body weight, blood pressure). I understand that it is not the objective of the work, but it could be enriched if the association between the differences in absolute levels and / or z-scores between methods, and the demographic, anthropometric and / or clinical variables of young people were analyzed.

Reviewer #3: The paper addresses a topic of actual relevance and contributes to answer current questions.

The work is clearly written. The methodological approach is adequately explained.

There are some issues, mostly methodological that should be considered and/or addressed.

- Data about characteristics of the studied population is scarce (e.g. information about hemodynamic conditions, cardiovascular risk factors prevalence was not given). What kind of exercise did the subjects practice?

- Why only data from 200 subjects were considered for the comparative analysis?

- Why did the authors choose to compare the approaches using t-tests? Why tools like Bland and Altman tests were not used to assess the agreement between methods? Did the differences between the methods show dependence on the height level (proportional errors)?

- This reviewer considers that it would be of value to analyze the equivalence between the allometric methods themselves and to compare their concordance with the reference methods. The authors state that there would be differences between allometric methods, with the specific allometric exponents as the preferred method. This issue and its significance in clinical practice should be accurately analyzed and discussed.

- Information about cofactors was not given. Were they considered? If so, was their impact on the differences methodological approaches similar? How did the authors defined covariates should not be considered in the analysis?

- How could the characteristics of the practiced sport have an impact on the results obtained and on the possibility of considering them in the usual clinical practice and in other population groups?

- The following sentence should be clarified: It must be noted that, because of a lack of a gold standard procedure, any conclusions based on the comparisons between different methods are limited.

- The tables design should be improved.

6. PLOS authors have the option to publish the peer review history of their article (what does this mean?). If published, this will include your full peer review and any attached files.

Reviewer #1: No

Reviewer #2: Yes: Daniel Bia

Reviewer #3: Yes: Yanina Zócalo

---

## [Author Response · Author response to Decision Letter 0]

25 Sep 2019

Response to reviewers 

Reviewer #1: I have evaluated with interest the manuscript "Left ventricular mass normalization for body size in children based on an allometrically adjusted ratio is as accurate as normalization based on the centile curves method.

Q1: The authors mistakenly use the guidelines developed for adults as a bibliographic reference (Lang 2015) without considering the recommendations for children and adolescents (Lopez 2010, Lopez 2018). It is known that in pediatrics specific percentiles based on sex and age should be used to know the evolution of growth. The authors provide an approach to normalize ventricular mass and avoid standardization based on percentiles but do not fail to show the advantages of this approach.

Thank you for reminding us of the 2010 recommendations of Lopez et al. We are familiar with this report and appreciate it. We agree that it should be cited and it will be included as a bibliographic reference. However, we cannot agree that the reference to the 2015 recommendation of Lang et al. is a mistake. Certainly, it depends on the context; we believe that in this case, it was appropriately used. Please note that in both the mentioned works of Lopez et al., the Lang's guideline (the 2005 version) was cited in 1-2 position.

Reviewer #2: The article addresses a very relevant topic for the area of pediatric medicine (cardiovascular area). The writing is very clear, and the analyzes performed are explained in detail. The authors are clearly aware of the problem analyzed. I have some suggestions and requirements that I believe can increase the quality of the manuscript.

Q2: Page 6: "Three sets of LVM-for-height normative data were developed based on each group’s records, using three different methods of normalization". From which table, database and/or web were the values of LVM (M), coefficient of variation (S), and skewness (L) for each height level obtained, to calculate the z-score for each child? Please be specific. How were the "expected, normative or reference" mean values and standar deviation levels obtained (calculated), to calculate the z-score by the allometrically adjusted ratio method (two variants)? Considering that PlosOne is a Journal read by professionals from a wide range of disciplines, it would be important to give specific examples of these calculations, indicating where the values used come from (e.g. please consider including this in an Annex).

Thank you for the comment and recommendation. The cited text is from the Methods section; it refers to the procedure of generating the L, M and S values. However, at the beginning of the sub-section “The LVM normative data for the mutual comparison”, in the Results section, we provide information about supplementary files (S1 and S2 Datasets) with the L, M and S values which were generated based on our data. Quote: “Two sets of LVM-for-height normative data, separate for girls and boys and generated based on the LMS method, are provided as L, M, and S values in supplementary text files (S1 and S2 Datasets, respectively).”

Additionally, in the sub-section “Development of left ventricular mass normative data”, we have added information about the data needed to generate the L, M and S values (quote): “In this method, based on the relationship between LVM and height in the study group, the expected mean LVM (M), coefficient of variation (S), and skewness (L) for each height level are generated. The LVM z-score is then calculated for an individual child, from the L, M, and S values corresponding to the child’s height, according to the equation:”

To add information on how the mean and standard deviation, recognized as the normative data, were obtained, we added text in the aforementioned section (quote): “The sex-specific allometric exponents were used to transform height, which is used as a denominator in the ratio method. Then, for each subject, LVM was divided by transformed height. Thus, new variables of indexed LVM were produced, and normative data expressed as a mean and standard deviation of the LVM indexes developed. Next, LVM z-scores were calculated, according to the equation:”

According to your suggestion, we have prepared a supplementary file with examples of LVM z-score calculations. We added a sentence to the manuscript (quote): “Examples of LVM z-score calculations are presented in a supplementary file (S1 Text)”

In the context of the comments and the queries, it seems important to note that the aim of our study is not a presentation of LVM normative data for a specific group of children and adolescents. It is a continuation of our previous work on improvement to the methodology of LVM scaling. In this study, we wanted to test if there is a significant superiority of the advanced methods (centile curves) over the practical approaches to LVM scaling.

Q3. Page 8: "For proper normalization of LVM for body size, it is necessary to eliminate body size information from the normalized LVM [20]. To check whether the body size information had been eliminated in the produced normative data, we tested whether there was a relationship between the calculated LVM z-scores and height. The Pearson correlation coefficient and the slope of the linear regression line for each set of the LVM z-scores were examined". Why were only linear models used for this purpose?

Good point. The procedure to verify whether normalization eliminates size information from measurement variables was described by Albrecht GH et al. (1993) and is widely accepted. They listed three equivalent criteria:

(1) Statistical - correlation coefficient (R) between the normalized variable and normalizing body size variable is zero or nearly so.

(2) Graphical - the least-squares regression line of the relationship between the normalized variable and normalizing body size variable has a slope of 0 (horizontal line on a scatterplot).

(3) Algebraic - valid for allometric methods only; the expected value of the normalized variable is equal to a constant.

However, it is possible that the correlation coefficient is close to 0, and the slope is 0, too, and there is a strong non-linear relationship. Therefore, Albrecht et al. have recommended inspecting graphical presentation of the data. In our study, the correlation coefficient and slope are both close to 0, and the arrangement of points on the scatterplots does not suggest a non-linear relationship. Of course, we are going to add the graphs to the supplementary data (S1 Figure).

Q4: Page 8: "Comparison of different methods of the LVM normalization: In this part of the study, from each of the sex-specific Study Group, 200 subjects were randomly assigned to corresponding Test Groups, to compare different LVM normalization methods". Why did you work with a subsample in this part of the analysis? As an example, did you consider using Bootstrapping?

For analysis of the agreement between tools or methods, the observations should be collected consecutively or randomly. The LVM z-scores in the Study Group do not meet this requirement. If the LVM z-scores in the Study Group had been calculated for the normative data developed based on the same group, the z-scores thus computed would have had a mean of 0, or very close to 0, and a standard deviation of 1, or very close to 1. Such a sample should be considered distorted. 

In the Study Limitations, we admitted that there is a lack of a distinct group for the assessment of the agreement, and that this would be the best option. Therefore, we decided to randomly select observations from the Study Group to develop a sample for the analysis of the reproducibility. 

The sample size necessary to perform the analysis significantly exceeds the requirement for such a study, which was estimated at 65 bi-variate measurements. We tripled that number. We did not consider the use of bootstrapping because the adopted procedure was statistically valid and sufficient.

Q5: Page 8: "The z-scores calculated based on the LVM normative data obtained according to the allometrically adjusted ratio methods were compared to those calculated based on the L, M, and S values from the LMS method [19]. This allowed us to evaluate the reproducibility of the allometric methods and to assess their sensitivity and specificity compared to the LMS method". Why not also use a Bland and Altman analysis for repeated samples? Do the differences between methods remain the same for any level of body-height (proportional errors)? It would be important to include this (very simple) analysis, which although it has limitations, is widely used to analyze the agreement between tools. Please add it.

We did not use the Bland-Altman plot because it has a critical limitation when the measurement’s range is from negative values, through zero, to positive values, and the outer tails of the data distributions that are compared are divergent; this is the case with our data.

Please see Figure 1 below. It shows scatterplot of the LMS z-scores (blue) and specific allometric z-scores (red) against height. Estimated quadratic curves are superimposed to illustrate the issue (note: they are not significant.). The curves are divergent on the outer ends. This means that the differences between the z-scores are greater there. Both, the LMS z-scores and the allometric z-scores contribute to the divergence because, for both, extreme data points have a big impact on the outer tails of the distribution.

Figure 1. (figure available in "Response to Reviewers" file attached to the re-submission)

When this situation occurs, the distribution of the differences is not normal (Figure 2). The Bland-Altman analysis requires normal distribution. The measurement variables themselves need not be normally distributed, but their differences should be.

Figure 2. (figure available in "Response to Reviewers" file attached to the re-submission)

According to the general idea of Bland-Altman analysis, if the assumption of a normal distribution is not met, data may be logarithmically transformed. In the presented case, they cannot. This is because the z-scores are positive and negative. Besides, an analysis directed to a search for the proportional error is not practicable. The Bland-Altman plot shows a biphasic configuration of data points and even a parabolic shape. The range of the measurement also limits ratio calculation; there is a risk of dividing by 0. The ratios estimated for z-scores close to 0 might be immense, and may highly influence the calculation, causing misinterpretation of the result.

Please see Figure 3, based on our data, illustrating the problem. Onto the classic Bland-Altman plot prepared for boys (the differences in green), the specific allometric z-scores (red) and the LMS z-scores (blue) have been imposed.

Figure 3. (figure available in "Response to Reviewers" file attached to the re-submission)

The mean difference is 0.0013. The upper and lower limits of agreement are 0.2785 and -0.2759, respectively (densely dashed lines). However, there is a specific, parabolic arrangement of the points with a clear upper border. The differences increase (in absolute value), going down both to the right and to the left of 0, but one cannot work with ratios to assess whether the differences increase proportionally to the magnitude of the measurement. This is because the numbers near 0 give abnormal results and there is a risk of dividing by 0. The slope of the line fitted to the raw data is 0 (solid line).

The results of the agreement analysis performed in our study, using the concordance correlation coefficient, show that the bias correction factors are equal to 1. Scale shifts and location shifts are minimal. There is no systemic and proportional bias.

We added the information about the Bland-Altman method to the Study Limitations: “The Bland-Altman method is the most commonly used method to measure agreement [35] and a question may arise why we did not use this method. The Bland-Altman plot is not suitable for our data; the plot has a limitation when the outer tails of the compared data distributions are divergent. In our data, the extreme data points of both the LMS z-scores and the allometric z-scores contribute to such divergence. As a result, the differences between the z-scores are greater on the outer ends. When this situation occurs, the distribution of the differences is not normal. The Bland-Altman analysis requires normal distribution; the measurement variables need not be normally distributed, yet their differences should be. An additional limitation is the measurement range, from negative, through zero, to positive. According to the general idea of Bland-Altman analysis, if the assumption of normality is not met, data may be logarithmically transformed. In the presented case, they cannot, because the z-scores are positive and negative. Besides, an analysis directed to a search for the proportional bias is not practicable. The Bland-Altman plot shows a biphasic configuration of data points and even a parabolic shape. The range of the measurement also limits ratio calculation: there is a risk of dividing by 0. The ratios estimated for z-scores close to 0 might be immense, and may highly influence the calculation, causing misinterpretation of the results.”

Q6: Page 9: "For this analysis, the subjects in the Test Groups were classified as having LVH when their z-score>1.65". Add reference

Sensitivity and specificity analysis is a test exercise in this study, and the cut-off to define LVH can be arbitrary. However, the z-score of 1.65, which is equivalent to the 95th percentile, has been widely used. As a reference, we will add the original article from 2008 by Bethany Foster et al., in which the authors discuss this issue extensively.

Q7: Results (page 10): The authors mentioned in the methodology: " This was a retrospective study based on data derived during periodic medical evaluation of child and adolescent athletes". Table 1. Please include more information on the subjects, in order to know to what extent their values are within "normality" (healthy expeted values). Information on the clinical and / or hemodynamic characteristics (blood pressure, heart rate, other echocardiographic data) of the subjects should be included. In addition, please include the minimum, median and maximum value of each variable. It is important to know the levels of the variables for which the study results are applicable.

Of course, we will provide additional data. The minimum, median and maximum value of each variable will be presented in Table 1 instead of mean and standard deviation. However, we want to emphasize again that the aim of our study is not a presentation of LVM normative data for a specific group of children and adolescents.

Q8: Results: The authors mentioned in the methodology: "The athletes in whom echocardiography revealed significant acquired or congenital heart diseases, affecting normal heart size and hemodynamics, were not included in the study". It would be interesting to know the results of a similar analysis performed in this subgroup of children and adolescents. You can do it? In this way it could be known to what extent the methods have similarity in these special cases.

We absolutely agree that it would be interesting. However, the vast majority of the youth athletes examined in our Center are healthy girls and boys. The number of excluded athletes is too small to perform such analysis.

Q9: Results: "The Pearson correlation coefficients and the slopes of the linear regression lines of the relationships between the calculated LVM z-scores and heights are also presented in Table 2". Please, can you show these graphics?

Yes; as mentioned, we will attach the graphs to the supplementary data (Figure S1).

Q10: Results: Table 5 and Table S2. Only 10 cases (in 200) were "positive" for LVH. Do you consider this "n" appropriate for a sensitivity and specificity analysis? Can you report confidence intervals? Can you present this analysis for "the whole group" (for separate and non-separated sexes).

Thank you for this question. It forced us to rethink the approach to analyzing sensitivity and specificity in terms of sample size. For expected high sensitivity and specificity, assuming a margin of error of 0.05, and considering the prevalence of LVH, the estimated sample size is about 760. Thus, we will analyze the entire group of 791 children and adolescents and introduce this change to the article. According to your request, we present tables with analysis for the whole group and for the separate sex-specific groups (below).

The whole group heightb height2.7

Sample size 791 791

Number of true positives 31 30

Number of true negatives 751 751

Number of false positives 9 9

Number of false negatives 0 1

Sensitivity 100.00% 96.77% 

95% Confidence Interval for Sensitivity 88.78% - 100.00% 83.30% - 99.92%

Specificity 98.82% 98.82 %

95% Confidence Interval for Specificity 97.76% - 99.46% 97.76% - 99.46%

Girls heightb height2.7

Sample size 327 327

Number of true positives 16 15

Number of true negatives 311 311

Number of false positives 0 0

Number of false negatives 0 1

Sensitivity 100% 93.75%

95% Confidence Interval for Sensitivity 79.41% - 100.00% 69.77% - 99.84%

Specificity 100% 100%

95% Confidence Interval for Specificity 98.82% - 100.00% 98.82% - 100.00%

Boys heightb height2.7

Sample size 464 464

Number of true positives 15 15

Number of true negatives 440 440

Number of false positives 9 9

Number of false negatives 0 0

Sensitivity 100% 100%

95% Confidence Interval for Sensitivity 78.20% - 100.00% 78.20% - 100.00%

Specificity 98.00% 98.00%

95% Confidence Interval for Specificity 96.23% - 99.08% 96.23% - 99.08%

Q11: Discussion (page 16): "It seems that there is not much difference between allometric normalization using a specific allometric exponent as compared to the universal allometric exponent. However, the exact numbers are slightly better for the specific exponents, and analysis of sensitivity and specificity also indicates the allometric normalization with the especific allometric exponents as the preferred method". An important question that arises is: Are children studied similar (in terms of LVM and body height) to children from other places. Could the authors perform an analysis and / or discuss this aspect? It is important for the purpose of understanding how generalizable the results are. Please consider comparing yourself to a population of different latitudes. An example may be the following article:

• Díaz A et al. Reference Intervals and Percentile Curves of Echocardiographic Left Ventricular Mass, Relative Wall Thickness and Ejection Fraction in Healthy Children and Adolescents. Pediatr Cardiol. 2019 Feb;40(2):283-301.

Are the children studied similar (in terms of LVM and height) to children from other places? To answer this question reliably we have to say no. Since anthropometric indices of children and adolescents depend on the economic and living conditions of a population, there are differences in height even between neighboring countries, with similar genetic backgrounds. The Dutch are the tallest in the world. Germans are slightly taller than Poles although, during a certain period of development, German boys are shorter than their Polish counterparts.

(Kułaga, Z., Litwin, M., Tkaczyk, M. et al. Polish 2010 growth references for school-aged children and adolescents. Eur J Pediatr (2011) 170: 599. https://doi.org/10.1007/s00431-010-1329-x)

To our knowledge, no study has compared absolute LVM in children from different countries. However, it is proven that malnutrition affects LVM.

(Di Gioia G, Creta A, Fittipaldi M, Giorgino R, Quintarelli F, Satriano U, et al. (2016) Effects of Malnutrition on Left Ventricular Mass in a North-Malagasy Children Population. PLoS ONE 11(5): e0154523. https://doi.org/10.1371/journal.pone.0154523).

What does the specificity of the group mean in the case of our study? The studied children were engaged in regular athletic training. Athletes are our group of interest. Regular exercise causes physiological changes to the heart, including hypertrophy. Therefore, a proper LVM assessment and differentiation of physiological LVH from pathological is important. An essential part of this assessment is reliable normalization of LVM for body size. It is particularly important in children and adolescents due to the large variability of body size in children of similar age. 

In our opinion, the group of child and adolescent athletes used in this study is representative of the population of the region. Potential differences in absolute LVM values compared to other regions with a similar economic situation are eliminated after normalization for body size.

In addition, we emphasize that it was not the aim of the present study to introduce LVM normative data for child and adolescent athletes. In previous studies, we presented normative data for child and adolescent athletes and compared them to that presented by others:

Krysztofiak H, Małek ŁA, Młyńczak M, Folga A, Braksator W (2018) Comparison of echocardiographic linear dimensions for male and female child and adolescent athletes with published pediatric normative data. PLoS ONE 13(10): e0205459. https://doi.org/10.1371/journal.pone.0205459

Krysztofiak H, Młyńczak M, Folga A, Braksator W, Małek ŁA. Normal Values for Left Ventricular Mass in Relation to Lean Body Mass in Child and Adolescent Athletes. Pediatr Cardiol (2019) 40: 204. https://doi.org/10.1007/s00246-018-1982-9

The aim of the study was just to verify whether it is necessary to use the sophisticated methodology to develop normative data for LVM. The key question in this study was: should we strive to replace the allometrically adjusted ratio method of LVM normalization with the more sophisticated method of centile curves in clinical practice? In the context of this question, the specificity of the group chosen for analysis does not affect the results. We are convinced that these results are universal and should be considered in clinical practice. 

As for the suggestion to compare our group with a population of different latitudes, for example, that studied by Diaz A et al (2019) - since our child and adolescent subjects are athletes, such comparison with the general population has limited rationale. However, we have found the recommended study relevant and will refer to it. It supports our idea of developing sex-specific rather than universal LVM normative data.

Q12: Discussion: An interesting aspect would be to know to what extent the differences (although apparently not significant) between methods, could be explained by other co-factors (e.g. sex, body weight, blood pressure). I understand that it is not the objective of the work, but it could be enriched if the association between the differences in absolute levels and / or z-scores between methods, and the demographic, anthropometric and / or clinical variables of young people were analyzed.

In general, we analyzed the agreement between two diagnostic methods. However, our methods differ from laboratory tests or medical equipment because they are based on the mathematical transformation of the same set of numbers - mathematical analysis of the same bivariate relationship. The final result, a z-score, is computed twice using different equations, but the initial value being transformed, absolute LVM, is the same. Co-factors influence the absolute LVM; they do not impact the equations the mathematical transformations. 

There is no significant difference between the computed z-scores. If there was a difference, it would be related not to co-factors, but to an error in the mathematical proceeding. It would be considered then as an error of the method. When trying to analyze the influence of the co-factors on the LVM, we analyze their impact on the absolute value of LVM, even if we test z-scores.

Simply put, co-factors like sex, body weight, and blood pressure affect the cardiac size that can be measured as linear dimensions in echocardiography. From the linear dimensions of the left ventricle, the LVM is calculated - this is the absolute value of LVM. At this point mathematical transformation starts that produces z-scores. In this study, we used three different mathematical transformations to produce z-scores. Co-factors do not impact the mathematical transformation.

This study used the same sample as a previous work, where we thoroughly discussed the effect of body weight on LVM:

Krysztofiak H, Młyńczak M, Małek ŁA, Folga A, Braksator W (2019) Left ventricular mass is underestimated in overweight children because of incorrect body size variable chosen for normalization. PLoS ONE 14(5): e0217637. https://doi.org/10.1371/journal.pone.0217637

As for differences in LVM related to sex, it is a very interesting topic. We will study this in upcoming research.

Reviewer #3: The paper addresses a topic of actual relevance and contributes to answer current questions. The work is clearly written. The methodological approach is adequately explained. There are some issues, mostly methodological that should be considered and/or addressed.

Q13: - Data about characteristics of the studied population is scarce (e.g. information about hemodynamic conditions, cardiovascular risk factors prevalence was not given). What kind of exercise did the subjects practice?

We will provide additional data including information about the volume of physical activity. All of the studied children were engaged in regular athletic training at the local or national level (mainly soccer, track and field, basketball, swimming, and martial arts).

Regarding cardiovascular risk factors, there is no such information because the subjects were healthy children and adolescents without cardiovascular risk factors like hyperlipidemia, diabetes, hypertension, etc. The only potential risk factor that is present in this group, in low prevalence, is overweight or obesity.

In the context of the comments and queries, it seems important to note that the aim of our study is not a presentation of LVM normative data for a specific group of children and adolescents. It is a continuation of our previous work on improvement of the methodology of LVM scaling. In this study, we wanted to test if there is a significant superiority of the advanced methods (centile curves) over the practical approaches to the LVM scaling.

Q14: - Why only data from 200 subjects were considered for the comparative analysis?

For analysis of the agreement between tools or methods, the observations should be collected consecutively or randomly. The LVM z-scores in the Study Group do not meet this requirement. If LVM z-scores in the Study Group had been calculated on the normative data developed based on the same group, the s-score variables thus computed would have had a mean very close to 0, and a standard deviation very close to 1. Such a sample should be considered distorted. 

In the Study Limitations, we admitted that there is a lack of a distinct group for the assessment of the agreement, and this would be the best option. Therefore, we decided to randomly select observations from the Study Group to develop a sample for analysis of the reproducibility. 

The sample size necessary to perform the analysis significantly exceeds the requirement for such a study, which was estimated at 65 bivariate measurements. We tripled that number.

Q15: - Why did the authors choose to compare the approaches using t-tests? Why tools like Bland and Altman tests were not used to assess the agreement between methods? Did the differences between the methods show dependence on the height level (proportional errors)?

The t-test used was a preliminary analysis. If the differences had been significant, this would have meant that no further study is needed. We would conclude that there is no agreement between the methods.

We did not use the Bland-Altman plot because it has a critical limitation when the measurement’s range is from negative values, through zero, to positive values, and the outer tails of the data distributions that are compared are divergent; this is the case with our data.

Please see Figure 1 below. It shows scatterplot of the LMS z-scores (blue) and specific allometric z-scores (red) against height. Estimated quadratic curves are superimposed, to illustrate the issue (note: they are not significant.). The curves are divergent on the outer ends. This means that the differences between the z-scores are greater there. Both, the LMS z-scores and the allometric z-scores contribute to the divergence because, for both, extreme data points have a big impact on the outer tails of the distribution.

Figure 1. (figure available in "Response to Reviewers" file attached to the re-submission)

When this situation occurs, the distribution of the differences is not normal (Figure 2). The Bland-Altman analysis requires normal distribution. The measurement variables themselves need not be normally distributed, but their differences should be.

Figure 2. (figure available in "Response to Reviewers" file attached to the re-submission)

According to the general idea of Bland-Altman analysis, if the assumption of a normal distribution is not met, data may be logarithmically transformed. In the presented case, they cannot. This is because the z-scores are positive and negative. Besides, an analysis directed to a search for the proportional error is not practicable. The Bland-Altman plot shows a biphasic configuration of data points and even a parabolic shape. The range of the measurement also limits ratio calculation; there is a risk of dividing by 0. The ratios estimated for z-scores close to 0 might be immense, and may highly influence the calculation, causing misinterpretation of the result.

Please see Figure 3, based on our data, illustrating the problem. Onto the classic Bland-Altman plot prepared for boys (the differences in green), the specific allometric z-scores (red) and the LMS z-scores (blue) have been imposed.

Figure 3. (figure available in "Response to Reviewers" file attached to the re-submission)

The mean difference is 0.0013. The upper and lower limits of agreement are 0.2785 and -0.2759, respectively (densely dashed lines). However, there is a specific, parabolic arrangement of the points with a clear upper border. The differences increase (in absolute value), going down both to the right and to the left of 0, but one cannot work with ratios to assess whether the differences increase proportionally to the magnitude of the measurement. This is because the numbers near 0 give abnormal results and there is a risk of dividing by 0. The slope of the line fitted to the raw data is 0 (solid line).

The results of the agreement analysis performed in our study, using the concordance correlation coefficient, show that the bias correction factors are equal to 1. Scale shifts and location shifts are minimal. There is no systemic and proportional bias.

We added the information about the Bland-Altman method to the Study Limitations: “The Bland-Altman method is the most commonly used method to measure agreement [35] and a question may arise why we did not use this method. The Bland-Altman plot is not suitable for our data; the plot has a limitation when the outer tails of the compared data distributions are divergent. In our data, the extreme data points of both the LMS z-scores and the allometric z-scores contribute to such divergence. As a result, the differences between the z-scores are greater on the outer ends. When this situation occurs, the distribution of the differences is not normal. The Bland-Altman analysis requires normal distribution; the measurement variables need not be normally distributed, yet their differences should be. An additional limitation is the measurement range, from negative, through zero, to positive. According to the general idea of Bland-Altman analysis, if the assumption of normality is not met, data may be logarithmically transformed. In the presented case, they cannot, because the z-scores are positive and negative. Besides, an analysis directed to a search for the proportional bias is not practicable. The Bland-Altman plot shows a biphasic configuration of data points and even a parabolic shape. The range of the measurement also limits ratio calculation: there is a risk of dividing by 0. The ratios estimated for z-scores close to 0 might be immense, and may highly influence the calculation, causing misinterpretation of the results.”

Q16: - This reviewer considers that it would be of value to analyze the equivalence between the allometric methods themselves and to compare their concordance with the reference methods. The authors state that there would be differences between allometric methods, with the specific allometric exponents as the preferred method. This issue and its significance in clinical practice should be accurately analyzed and discussed.

As stated in the submitted article, it was not the aim of our study to question the exponent of 2.7, so we did not make a direct comparison between this variant and specific variants. However, since our study group consisted of child and adolescents athletes, we were interested in whether the application of specific allometric exponents would improve the performance of the allometric method when testing against the centile curves method of LVM normalization. Parallel analysis with two variants of the allometric method, with the universal exponent and with exponents specific to young athletes, has let us estimate the performance of both variants. 

We did not state that there is a difference between allometric methods. We wrote that the specific variant seems to work better, but that the universal variant is almost equally effective. Although the exact estimates of reproducibility indicate slightly better agreement in the case of the specific allometric exponents, the graphs and the concordance coefficients reflect high precision and accuracy, with minimum location shift and scale shift, for both the allometric variants.

However, because of better sensitivity in our study, we have stated that the analysis of sensitivity and specificity indicates the allometric normalization with the specific allometric exponents as the preferred method. In the discussion, we noted that the universal exponent of 2.7 has been questioned by others. As a major drawback, researchers point to the presence of a relationship between normalized LVM and height. Therefore, we wrote in the conclusion that it seems that group-specific allometric exponents should be used to avoid constraints related to incomplete elimination of body size information from the normalized LVM, and for better performance in daily clinical practice.

Below is a table with concordance correlation coefficients. The rightmost column shows the results of comparing the allometric methods. The results are predictable because we know the reason why the agreement is so close to perfection, yet not perfect: there is a minimal difference between the allometric exponents.

 heightb vs. LMS height2.7 vs. LMS heightb vs. height2.7

Girls 

Pearson correlation coefficient 0.9917 (p<0.001) 0.9886 (p<0.001) 0.9974 (p<0.001)

Bias correction factor 1.0000 1.0000 1.0000

Concordance correlation coefficient 0.9917 0.9886 0.9974

Lower one-sided 95% CI 0.9895 0.9857 0.9968

Scale shift 1.0001 1.0031 1.0030

Location shift 0.0043 0.0041 0.0002

Boys 

Pearson correlation coefficient 0.9916 (p<0.001) 0.9870 (p<0.001) 0.9974 (p<0.001)

Bias correction factor 1.0000 0.9999 1.0000

Concordance correlation coefficient 0.9916 0.9869 0.9974

Lower one-sided 95% CI 0.9894 0.9834 0.9967

Scale shift 0.9958 0.9954 0.9997

Location shift 0.0067 0.0160 0.0092

Q17: - Information about cofactors was not given. Were they considered? If so, was their impact on the differences methodological approaches similar? How did the authors defined covariates should not be considered in the analysis?

In general, we performed an analysis of the agreement between two diagnostic methods. However, our methods differ from laboratory tests or medical equipment because they are based on mathematical transformation of the same set of numbers - mathematical analysis of the same bivariate relationship. The final result, a z-score, is computed twice using different equations, but the initial value being transformed, absolute LVM, is the same. Co-factors influence the absolute LVM, not the equations or the transformation. 

There is no significant difference between the computed z-scores. If there was a difference, it would not be related to co-factors, but to an error in the mathematical process. This would be an error of the method. When we are trying to analyze the influence of the co-factors on the LVM, we assess their impact on the absolute value of LVM, even if we test z-scores.

Simply put, co-factors like sex, body weight, and blood pressure affect the cardiac size that can be measured as linear dimensions in echocardiography. From the linear dimensions of the left ventricle, the LVM is calculated - this is the absolute value of LVM. At this point, mathematical transformation starts that produces z-scores; we used three different mathematical transformations to produce z-scores. Co-factors do not impact the mathematical transformation.

Q18: - How could the characteristics of the practiced sport have an impact on the results obtained and on the possibility of considering them in the usual clinical practice and in other population groups?

The key question in this study was: should we strive to replace the allometrically adjusted ratio method of LVM normalization with the more sophisticated method of centile curves in clinical practice? In the context of this question, the specificity of the group, we chose to analyze does not have an impact on the results. Yet there was a secondary question: does a specific population need a specific allometric exponent to develop normative LVM data with an allometrically adjusted ratio? In our case, the population was represented by a group of young athletes. In this part, the results show some difference, thus, the specificity of the group had an impact on the results. 

However, you have posed a deeper question about the characteristics of the practiced sport and its impact on the results. We are convinced that the characteristics of the practiced sports in our group did not affect the general results. The results of the study should be considered in clinical practice. The studied children were engaged in regular athletic training at the local or national level, mainly soccer, track and field, basketball, swimming, and martial arts. At this stage of athletic development, training is primarily focused on the systematic development of motor abilities. The profile of the group reflects the population in the context of a practiced sport. In our opinion, the group of child and adolescent athletes used in this study is representative of the population of the region.

Q19: - The following sentence should be clarified: It must be noted that, because of a lack of a gold standard procedure, any conclusions based on the comparisons between different methods are limited.

In our study, we regarded the LMS method as a reference, a current standard, the most accurate procedure available. When we use the term 'gold standard test', we mean a method that is able to determine or exclude disease with certainty. In LVM normalization, this certainty is not possible because this is not a direct measurement and many confounding factors impact the result. Therefore, the best standard should be established after careful consideration of as many limiting factors as possible.

The sentence "It must be noted that, because of a lack of a gold standard procedure, any conclusions based on the comparisons between different methods are limited" is used in a paragraph discussing a comparison of different methods of LVM normalization, with different body size scaling variables. We are trying to draw attention to methodological drawbacks when we arbitrarily assume that one of the methods is the most accurate. We are aware that the LMS method also has some limitations (for example, as stated in the WHO document, extreme data points have a big impact on the outer tails of the distribution) and that we should take them into account when formulating conclusions favoring one method, and disqualifying another.

However, we agree that the sentence should be modified for the sake of clarity. We made the following change (quote): " It must be noted that because of the lack of a procedure that is able to determine or exclude LV hypertrophy with certainty, any conclusions based on the comparisons between different methods, favoring one method or disqualifying another, are questionable."

Q20: - The tables design should be improved.

Thank you, we will improve the design of the tables.

---

## [Editor Report · Decision Letter 1]

1 Nov 2019

Left ventricular mass normalization for body size in children based on an allometrically adjusted ratio is as accurate as normalization based on the centile curves method.

PONE-D-19-17289R1

Dear Dr. Hubert Krysztofiak,

We are pleased to inform you that your manuscript has been judged scientifically suitable for publication and will be formally accepted for publication once it complies with all outstanding technical requirements.

With kind regards,

Alejandro Diaz

Guest Editor

PLOS ONE

The answers have been correctly argued

This version can be accepted for publication

---

## [Editor Report · Acceptance letter]

12 Nov 2019

PONE-D-19-17289R1 

Left ventricular mass normalization for body size in children based on an allometrically adjusted ratio is as accurate as normalization based on the centile curves method. 

Dear Dr. Krysztofiak:

I am pleased to inform you that your manuscript has been deemed suitable for publication in PLOS ONE. Congratulations! Your manuscript is now with our production department. 

With kind regards,

on behalf of

MD PhD Alejandro Diaz 

Guest Editor

PLOS ONE